# Old can be Gold: Better Gradient Flow can Make Vanilla-GCNs Great Again

**Ajay Jaiswal\*, Peihao Wang\*, Tianlong Chen, Justin F. Rousseau, Ying Ding, Zhangyang Wang**
University of Texas at Austin
{ajayjaiswal, peihaowang, tianlong.chen, atlaswang}@utexas.edu
{justin.rousseau, ying.ding}@austin.utexas.edu

## Abstract

Despite the enormous success of Graph Convolutional Networks (GCNs) in modeling graph-structured data, most of the current GCNs are shallow due to the notoriously challenging problems of over-smoothening and information squashing along with conventional difficulty caused by vanishing gradients and over-fitting. Previous works have been primarily focused on the study of over-smoothening and over-squashing phenomena in training deep GCNs. Surprisingly, in comparison with CNNs/RNNs, very limited attention has been given to understanding how healthy gradient flow can benefit the trainability of deep GCNs. In this paper, firstly, we provide a new perspective of **gradient flow** to understand the substandard performance of deep GCNs and hypothesize that by facilitating *healthy gradient flow*, we can significantly improve their trainability, as well as achieve state-of-the-art (SOTA) level performance from vanilla-GCNs [1]. Next, we argue that blindly adopting the Glorot initialization for GCNs is not optimal, and derive a **topology-aware isometric initialization** scheme for vanilla-GCNs based on the principles of isometry. Additionally, contrary to ad-hoc addition of skip-connections, we propose to use **gradient-guided dynamic rewiring of vanilla-GCNs** with skip connections. Our dynamic rewiring method uses the gradient flow within each layer during training to introduce *on-demand* skip-connections adaptively. We provide extensive empirical evidence across multiple datasets that our methods improve gradient flow in deep vanilla-GCNs and significantly boost their performance to *comfortably compete and outperform* many fancy state-of-the-art methods. Codes are available at: `https://github.com/VITA-Group/GradientGCN`.

## 1 Introduction

Graphs are omnipresent and Graphs convolutional networks (GCNs)[1] and their variants[2, 3, 4, 5, 6, 7, 8, 9, 10] are a powerful family of neural networks which can learn from graph-structured data. GCNs have been enormously successful in numerous real-world applications such as recommendation systems [11, 12], social and academic networks[13, 11, 5, 4, 14], modeling proteins for drug discovery[15, 16, 12], computer vision[17, 18], etc. Despite their popularity, training deep GCNs are notoriously hard and many classical GCNs[1, 3, 19] achieve their best performance with shallow depth and completely bilk trainability with increasing stacks of layers and non-linearity[20, 21].

Previous works have focused on studying the roadblocks of training deep GCNs from the perspective of *over-smoothening* [20, 21] and *information bottleneck* [22] and many approaches broadly categorized as *architectural* [23, 24, 20, 25], and *regularization & normalization* [26, 27, 28, 29] tweaks has been proposed for mitigation. In recent work, [30] theoretically validated that the deeper GCN model is at least as expressive as the shallow GCN model, as long as deeper GCNs are trained properly.

---

\*Equal Contribution.

36th Conference on Neural Information Processing Systems (NeurIPS 2022).

Furthermore, [31] studied the training difficulty of GCNs from the perspective of graph signal energy loss, and proposed modifying the GCN operator, from the energy perspective but it failed to recover the full potential of vanilla-GCNs. Surprisingly, unlike traditional neural networks (CNNs and RNNs), limited effort has been given to understand hurdles in the trainability of deep GCNs from the signal propagation perspective, i.e. **gradient flow**. In this paper, we comprehensively show that deep GCNs suffer from *poor gradient flow* (error signals) under back-propagation across multiple datasets and architectures, which significantly hurt their trainability and lead to substandard performance. Further, we are motivated to explore an orthogonal direction: *Can we make vanilla-GCNs [1] **go deeper** and **comparable/better** than SOTA, by improvising healthy gradient flow during training?*

In this work, we *firstly* look into the effect of initialization (surprisingly overlooked) in GCNs, and find that blindly adopting the Glorot initialization [32] from CNNs has critical impacts on the gradient flow of GCN training, especially for deep GCNs. When the initial weights are not chosen appropriately, the propagation of the input signal into the layers of random weights can result in poor error signals under back-propagation [33]. Inspired from the signal propagation perspective, we leverage the *principle of isometry* [34, 35] to derive a *theoretical initialization scheme* for vanilla-GCNs, by incorporating the graph topological information - which yields a remarkably easy-to-compute form.

Skip-connections have been very resourceful to train deep convolutional networks [36, 37] by improving the gradient flow, and recently some works [23, 24, 38, 20] have identified their benefits for deep GCNs. Although their benefits are known, most of the existing works attempt to add skip-connections in an *ad-hoc and non-optimal* fashion in deep GCNs which creates additional overhead of storing multiple activations during training along with substandard performance. *Secondly*, we study layerwise gradient flow during the training of deep vanilla-GCNs and identify that there exist some layers which receive almost zero error signal (i.e. gradients) during backpropagation and lose trainability and expressiveness. We use *computationally efficient* Gradient Flow ($p$-norm of gradients within a layer) as a metric to dynamically identify such layers during training and propose a principled way of injecting skip-connections on-demand to facilitate healthy gradient flow in deep GCNs.

Our main contributions can be summarized as:

- **Deep GCNs suffer from poor gradient flow during training.** We rigorously investigate the *gradient flow* during the training of GCNs and observe that GCNs suffer from poor gradient flow which worsens with the increase in depth. We hypothesize that by improving the gradient flow in deep vanilla-GCNs, along with effectively improving their trainability, we can achieve SOTA-level performance from vanilla-GCNs [1].

- **Topology-Aware Isometric Initialization of GCNs.** Unlike blind adoption of Glorot initialization [32] for GCNs, we derive a *theoretical initialization scheme* for vanilla-GCNs [1] based on the principle of isometry, which relates benign initial parameters with graph topology. We experimentally validate that our new initialization strategy significantly improves the gradient flow in deep GCNs and provide huge performance benefits.

- **Gradient-Guided Dynamic Rewiring of GCNs.** Contrary to *ad-hoc* addition of skip-connections to improve GCNs performance, in this paper, we leverage Gradient Flow to introduce *dynamic rewiring strategy* of vanilla-GCNs with skip-connections. Our new rewiring improves gradient flow, reduces the overhead of storing multiple intermediate activations (dense, initial, residual, jumping skips), as well as outperforms many ad-hoc skip connection mechanisms.

## 2 Methodology

In this section, we aim to discuss the trainability of deep GCNs from the signal propagation perspective, i.e. gradient flow. We will provide a detailed introduction and theoretical derivation of our topology-aware isometric initialization. Furthermore, we will present a novel way to incorporate skip-connections using Gradient Flow to improve the trainability of deep GCNs.

### 2.1 Preliminaries

We begin by formulating Graph Convolutional Networks (GCN) along the way introducing our notation. Let $G = (\mathcal{V}, \mathcal{E})$ be a simple and undirected graph with vertex set $\mathcal{V}$ and edge set $\mathcal{E}$. Let

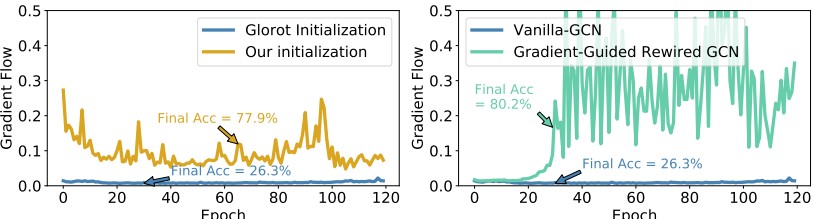

Figure 1: (a) Comparison of Gradient flow in 10-layer vanilla-GCN with default (Glorot initialization) and our Topology-Aware initialization. (b) Comparison of Gradient flow in 10-layer GCN and GCNs with our new Gradient Flow Guided Rewiring.

$A \in \mathbb{R}^{N \times N}$ be the associated adjacency matrix, such that $A_{ij} = 1$ if $(i, j) \in \mathcal{E}$, and $A_{ij} = 0$ otherwise. $N = |\mathcal{V}|$ is the number of nodes. We also define $d_i = \sum_j A_{ij}$ as the node degree of the $i$-th vertex. Let $X \in \mathbb{R}^{C \times N}$ be the node feature matrix, whose $i$-th column represents a $d$-dimension feature vector for the $i$-th node. GCNs aim to learn an embedding for each node via learnable graph convolutional filters [2, 1]. A graph convolutional layer can be formulated as [2]:

$$f(G, X) = W X (A + I),\tag{1}$$

where adding $I$ to adjacency matrix is known as the self-loop trick [1], $W \in \mathbb{R}^{C' \times C}$ is the weight matrix of this layer, and $C'$ is the dimension of output channel. An $L$-layer GCN cascades $L$ layer of graph convolutional layers followed by non-linear activations:

$$Y = f^{(L)} \circ \sigma \circ f^{(L-1)} \circ \cdots f^{(2)} \circ \sigma \circ f^{(1)}(G, X),\tag{2}$$

where $Y \in \mathbb{R}^{C'}$ is the final output of the GCN, and $\sigma(\cdot)$ denotes a non-linear activation, which is typically chosen as ReLU. Although there exists lots of message-passing based Graph Neural Networks (GNNs), which are incorporated with highly sophisticated mechanisms, such as attention [3] and gating [39, 40], the focus of our work only lies on the most classical GCN model (Eq. 2 or [1]) that exactly follows the rigorous definition of graph convolutions [41]. As we will show in our experiments, our proposal will bring this theoretically clean GCN back to the SOTA performance.

## 2.2 Understanding Gradient Flow in Deep GCNs

Gradient flow and its associated problems such as exploding and vanishing gradients have been extensively explored in the dynamics of learning of neural networks (CNNs/RNNs) [32, 33, 35]. GCNs [1] are a special category of neural networks, which uses "graph convolution" operation (linear transformation of all neighbors of a node followed by non-linear activation) to learn the graph representation. Despite being powerful in learning high-quality node representations, GCNs have limited ability to extract information from high-order neighbors and significantly lose their trainability and performance. Recently, many works [20, 21, 22, 24]

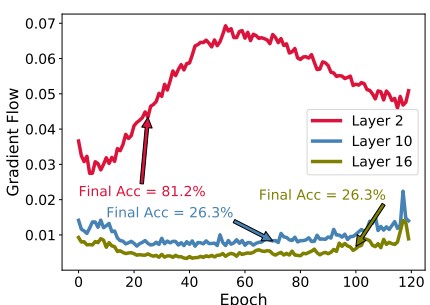

Figure 2: Gradient flow of vanilla-GCNs (layer 4,10,16) trained on Cora.

have been proposed to address this issue from the perspective of over-smoothing and information bottleneck. However, the *signal propagation* perspective to understand the trainability and performance of GCNs has been highly overlooked compared to the attention it has received for CNNs/RNNs.

Gradient flow is used to study the optimization dynamics of neural networks and it has been widely known that healthy gradient flow facilitates better optimization [32, 33, 35, 42]. Consider an $L$-layer GCN (see Eq. 2), we denote the weight parameter for the $l$-th layer $f^{(l)}$ as $W_l$. Suppose we have cost function $\mathcal{L}$, we calculate the gradient across each layer and effective gradient flow (GF) as:

$$g_1 = \frac{\partial \mathcal{L}}{\partial W_1}, \cdots, g_i = \frac{\partial \mathcal{L}}{\partial W_i}, \cdots, g_L = \frac{\partial \mathcal{L}}{\partial W_L}\tag{3}$$

$$\text{Gradient Flow: GF}_p = \frac{1}{L} \sum_{n=1}^{L} \|g_n\|_p\tag{4}$$

---

[2] Here we consider $A$ is symmetrically normalized by default.

where for every layer $l$, $\frac{\partial \mathcal{L}}{\partial \boldsymbol{W}_l}$ represents gradients of the learnable parameters $\boldsymbol{W}_l$ and we have used $p = 2$ in our experiments. Figure 2 represents the gradient flow during training of vanilla-GCNs with layer 4, 6, and 10 on the Cora dataset. Figure 3 illustrates the comparison of validation loss and gradient flow in vanilla-GCNs with 2 and 10 layers on Cora, Citeseer, and Pubmed. We consistently observed across each dataset that with increasing depth, the gradient flow across layers decreases significantly along with the performance. In this paper, we hypothesize that by improving the gradient flow in deep vanilla-GCNs, along with effectively improving their trainability, we can achieve SOTA-level performance from vanilla-GCNs. We propose a new topology-aware initialization strategy based on the principles of isometry and dynamic architecture rewiring for vanilla-GCNs to improvise the gradient flow and subsequently performance (Figure 1(a),(b)).

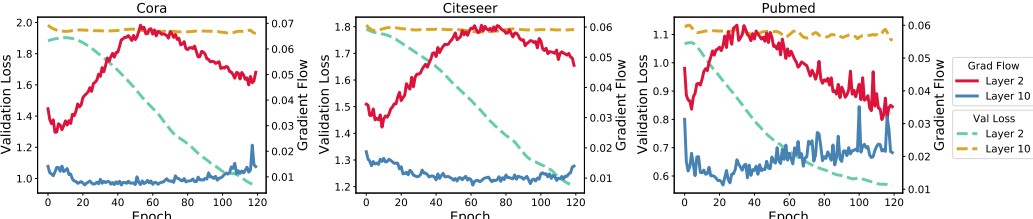

Figure 3: Comparison of validation loss and gradient flow in vanilla-GCNs with 2 and 10 layers.

## 2.3 Topology-Aware Isometric Initialization

Recent work of [43] has revealed that initializing network parameters nearly isometric can train very deep CNNs without any normalization. We are inspired to properly initialize each GCN layer to make it an isometric mapping. To begin with, we first give the formal definition of isometry below:

**Definition 1.** *[43] A map $f : \mathcal{X} \to \mathcal{Y}$ between two inner-product spaces is called an isometry if*

$$\langle f(\boldsymbol{x}), f(\boldsymbol{x'}) \rangle = \langle \boldsymbol{x}, \boldsymbol{x'} \rangle \tag{5}$$

*for all $\boldsymbol{x}, \boldsymbol{x'} \in \mathcal{X}$.*

Existing isometric initialization techniques are not free lunch for GCNs. Qi *et al.* [43] showed that a convolutional operator is isometric if and only if their cross-channel weights are orthogonal to each other. The straightforward implication is that delta kernel can be served as the isometric initialization. However, computing delta kernel in graph signal processing needs to involve complex numbers and eigen-decomposition of the adjacency matrix [44], which is infeasible for a common deep learning framework. To this end, we re-establish theoretical results for GCNs from Definition 1.

Consider one GCN layer (Eq. 1), our goal is to seek a weight matrix such that pair-wise angles between node features are invariant after transforming by the layer. First of all, we denote the output feature of the $i$-th node as $\boldsymbol{y}_i = \boldsymbol{W} \boldsymbol{X} \boldsymbol{a}_i$ where $\boldsymbol{a}_i$ is the $i$-th column of $(\boldsymbol{A} + \boldsymbol{I})$. The inner product of two node features $i, j \in [N]$ can be computed as:

$$\langle \boldsymbol{y}_i, \boldsymbol{y}_j \rangle = \left[ (\boldsymbol{a}_i^\top \otimes \boldsymbol{W}) \operatorname{vec}(\boldsymbol{X}) \right]^\top \left[ (\boldsymbol{a}_j^\top \otimes \boldsymbol{W}) \operatorname{vec}(\boldsymbol{X}) \right] \tag{6}$$

$$= \operatorname{Tr} \left( \left[ (\boldsymbol{a}_i \boldsymbol{a}_j^\top) \otimes (\boldsymbol{W}^\top \boldsymbol{W}) \right] \operatorname{vec}(\boldsymbol{X}) \operatorname{vec}(\boldsymbol{X})^\top \right), \tag{7}$$

where $\operatorname{vec}(\cdot)$ denotes vectorization of a matrix, Eq. 6 follows from the equality $\operatorname{vec}(\boldsymbol{ABC}) = (\boldsymbol{C}^\top \otimes \boldsymbol{A}) \operatorname{vec} \boldsymbol{B}$ [45]. On the other hand, $\boldsymbol{x}_i = \boldsymbol{I}_C \boldsymbol{X} \boldsymbol{e}_i$ where $\boldsymbol{e}_i$ is the $i$-th canonical basis, we can obtain a similar form of $\langle \boldsymbol{x}_i, \boldsymbol{x}_j \rangle$:

$$\langle \boldsymbol{x}_i, \boldsymbol{x}_j \rangle = \operatorname{Tr} \left( \left[ (\boldsymbol{e}_i \boldsymbol{e}_j^\top) \otimes \boldsymbol{I}_C \right] \operatorname{vec}(\boldsymbol{X}) \operatorname{vec}(\boldsymbol{X})^\top \right) \tag{8}$$

To achieve isometry, we hope to solve a $\boldsymbol{W}$ such that Eq. 8 equals to Eq. 7. However, a simple closed-form solution does not exist in general. Instead, we seek a $\boldsymbol{W}$ to minimize the difference between Eq. 8 and Eq. 7.

$$\boldsymbol{W}^* = \operatorname*{argmin}_{\boldsymbol{W} \in \mathbb{R}^{C' \times C}} \sum_{i,j \in [N]} \left| \langle \boldsymbol{y}_i, \boldsymbol{y}_j \rangle - \langle \boldsymbol{x}_i, \boldsymbol{x}_j \rangle \right|^2 \tag{9}$$

$$= \operatorname*{argmin}_{\boldsymbol{W} \in \mathbb{R}^{C' \times C}} \sum_{i,j \in [N]} \operatorname{Tr}^2 \left( \left[ (\boldsymbol{a}_i \boldsymbol{a}_j^\top) \otimes (\boldsymbol{W}^\top \boldsymbol{W}) - (\boldsymbol{e}_i \boldsymbol{e}_j^\top) \otimes \boldsymbol{I}_C \right] \operatorname{vec}(\boldsymbol{X}) \operatorname{vec}(\boldsymbol{X})^\top \right). \tag{10}$$

However, minimizing objective Eq. 10 will involve data matrix into the solution, which limits computational efficiency and generalization. Hence, we optimize a looser upper bound of Eq. 10 by minimizing the difference between $(\boldsymbol{a}_i \boldsymbol{a}_j^\top) \otimes (\boldsymbol{W}^\top \boldsymbol{W}$ and $(\boldsymbol{e}_i \boldsymbol{e}_j^\top) \otimes \boldsymbol{I}_C$. The intuition is that once this difference reaches zero, Eq. 10 will be minimized to zero as well.

$$\boldsymbol{W}^* = \underset{\boldsymbol{W} \in \mathbb{R}^{C' \times C}}{\operatorname{argmin}} \sum_{i,j \in [N]} \left\| \left(\boldsymbol{a}_i \boldsymbol{a}_j^\top\right) \otimes \left(\boldsymbol{W}^\top \boldsymbol{W}\right) - \left(\boldsymbol{e}_i \boldsymbol{e}_j^\top\right) \otimes \boldsymbol{I}_C \right\|_F^2. \tag{11}$$

At the first glance, Eq. 11 is non-convex and Kronecker product will produce a computational prohibitive matrix, which makes this problem intractable. However, it has not escaped our notice that the structure of $\boldsymbol{a}_i \boldsymbol{a}_j$ is highly sparse. Hence, the entire norm minimization can be decomposed into block-wise minimization problems. We defer the detailed derivation into Appendix **??**. As the consequence, the optimal solution to Eq. 11 should satisfy:

$$\|\boldsymbol{w}_k\|_2^2 = \frac{N^2}{\sum_{i,j \in [N]}(d_i + 1)(d_j + 1)} \qquad \forall k \in [C], \tag{12}$$

$$\boldsymbol{w}_k^\top \boldsymbol{w}_l = 0 \qquad \forall k, l \in [C], k \neq l, \tag{13}$$

where $\boldsymbol{w}_k$ denotes the $k$-th column of optimal weight matrix $\boldsymbol{W}^*$. Eq. 13 suggests that isometry requires weights for different channels to be orthogonal, which conforms with the general results of [43]. But different from [43], our results indicate that the magnitude of channel-wise weights should be constrained by a degree-related constant (Eq. 12), which binds the initial weights with topological information. To randomly initialize $\boldsymbol{W}$ in practice, one can draw each column of $\boldsymbol{W}$ from independent distributions (e.g., white Gaussian) with variance $\Sigma^2 = N^2 / \left[ C' \sum_{i,j}(d_i + 1)(d_j + 1) \right]$ and to initialize weights with bounded values, we suggest employ the following uniform distribution:

$$\boldsymbol{W} \sim \mathcal{U} \left[ -\sqrt{\frac{3N^2}{C' \left( \sum_{i,j \in [N]}(d_i + 1)(d_j + 1) \right)}}, \sqrt{\frac{3N^2}{C' \left( \sum_{i,j \in [N]}(d_i + 1)(d_j + 1) \right)}} \right] \tag{14}$$

### 2.4 Gradient-Guided Dynamic Rewiring

Graph Convolutions can be considered as a type of Laplacian smoothing, and repeatedly applying Laplacian smoothing many times in the case of multi-layer GCNs, leads the representations of the nodes in GCN to converge to a certain value and thus become indistinguishable, i.e. lose expressiveness and trainability [20, 46]. Despite numerous efforts from architectural, regularization, and normalization perspectives [23, 26, 27, 24, 20, 25, 28, 29], there still exists a wide gap in fully understanding the trainability issue of deep GCNs which can aid in developing techniques which can prevent performance deterioration with increasing depth.

In this work, we look for the effect of increasing depth on the error signal propagation during training and found that deep GCNs suffer from poor gradient flow and it worsens dramatically with increasing depth, thereby completely blocking the gradient propagation and potentially leading to a catastrophic failure to update during training. More specifically, we performed gradient flow analysis during training of deep vanilla-GCNs (Figure 4 (a)) and found that with the progress in training, many deep GCN layers receive almost zero error signal during backpropagation (i.e. gradients) and unable to train. Motivated by this finding, we use computationally efficient Gradient Flow as a metric to identify layers that block healthy error signal back-propagation during training and propose to dynamically rewire the vanilla-GCNs using skip-connections (widely known to improve gradient flow), which we call *on-demand dynamic rewiring*. Our dynamic rewiring technique *improves gradient flow, mitigate sudden feature collapse and significantly helps in training* deep vanilla-GCNs with high performance (Figure 1(b) and 4(c)). Mathematically, our dynamic rewiring can be written as:

$$\widetilde{\boldsymbol{X}}_t^l = \boldsymbol{W}_{l-1} \boldsymbol{X}_t^{(l-1)} (\boldsymbol{A} + \boldsymbol{I}) \tag{15}$$

$$\boldsymbol{X}_t^l = \mathbb{1}[\mathrm{GF}(\boldsymbol{X}_t^{(l)}) < p \cdot \mathrm{GF}(\boldsymbol{X}_0^{(l)})] \alpha \boldsymbol{X}_t^{(l-1)} + \widetilde{\boldsymbol{X}}_t^l \tag{16}$$

where, $\mathbb{1}[\cdot]$ is an indicator function, the subscript $t$ denotes the training epoch, the superscript $l$ denotes the layer index, $GF$ denotes gradient flow, $\alpha$ denotes skip information ratio, and $p$ denotes gradient flow drop threshold.

Precisely, we observe the Gradient Flow of each vanilla-GCN layer during training, and if the flow drop by $p\%$ (hyperparameter) of its initial value at the start of training, we assist the layer with a skip-connection. We use a modified version of *initial residual connection* [47, 24], which we define as the output feature matrix of first layer of vanilla-GCN (gives better performance than [47, 24]), to supplement the layers that quickly loses energy. Once the layers which are more prone to lose expressiveness are identified and rewired using the skip-connections, the vanilla-GCN training can proceed as normal.

In comparison with previous work [23, 24, 38, 20] which blindly inherits skip-connection techniques from CNNs/RNNs, our work *provides a principled approach to perform architectural rewiring* of vanilla-GCNs. Static skips introduced in [23, 24, 38, 20], by default incorporate skips for training shallow GCNs with 2-3 layers (which do not require it), and tend to hurt the final performance. For example, in Table 3, it can be observed that training a 2-layered GCN with initial, jumping, or residual skips suffers -2.1%, -0.12%, -6.37% performance drop compared to a plain vanilla-GCN (81.10%) on Cora. Our method only introduces skip-connections when it is required by tracking the gradient flow of the layer, and thereby overcomes the issue of adding skips in shallow GCNs, and for layers that receive healthy gradient flow. Our extensive experiments on multiple datasets reveal that our approach of guided rewiring not only brings benefits on the efficiency front (memory overhead to store intermediate activations) but also comfortably outperforms all SOTA ad-hoc skip-connection.

## 3 Experiment

In this section, we first provide experimental evidence to augment our signal propagation hypothesis and show that our newly proposed methods facilitate healthy gradient flow during the training of deep-vanilla GCNs. Next, we extensively evaluate our methods against state-of-the-art graph neural network models and techniques to improve vanilla-GCNs on on a wide variety of open graph datasets.

| Settings | Cora | Citeseer | Pubmed | OGBN-ArXiv |
|---|---|---|---|---|
| {Learning rate, Weight Decay, Hidden dimesnion} | $\{0.005, 5e-4, 64\}$ | $\{0.005, 5e-4, 64\}$ | $\{0.01, 5e-4, 64\}$ | $\{0.005, 0, 256\}$ |

Table 1: Hyperparameter configuration for our proposed method on representative datasets.

### 3.1 Dataset and Experimental Setup

We use three standard citation network datasets Cora, Citeseer, and Pubmed [48] in GNN domain for evaluating our proposed methods against state-of-the-art GNN models and techniques. We have used the basic vanilla-GCN implementation in PyTorch provided by the authors of [1] to incorporate our proposed techniques and show their effectiveness in making traditional GCN comparable/better with SOTA. For our evaluation on Cora, Citeseer, Pubmed, and OGBN-ArXiv, we have closely followed the data split settings and metrics reported by the recent benchmark [49]. See details in Appendix **??**. For comparison with SOTA models, we have used JKNet [50], InceptionGCN [51], SGC [52], GAT [3], GCNII [24], and DAGNN [53]. We use Adam optimizer for our experiments and performed a grid search to tune hyperparameters for our proposed methods and reported our settings in Table 1. For all our experiments, we have trained our modified GCNs for 1500 epochs and 100 independent repetitions following [49] and reported average performances with the standard deviations of the node classification accuracies. All experiments on large graph datasets, e.g., OGBN-ArXiv, are conducted on single 48G Quadro RTX 8000 GPU, while small graph experiments are completed using a single 16G RTX 5000 GPU.

### 3.2 Gradient Flow and our proposed methods

Despite significant efforts to improve deep neural networks (CNNs/RNNs) training from the signal propagation perspective, in-depth analysis of deep GCNs training with a focus on the gradient flow during back-propagation is highly overlooked. We use Equation 4 to study gradient flow during the training of GCNs and Figure 2 indicates that with an increase in depth, gradient flow in the network drop significantly hurting the trainability of GCNs. Figure 3 indicates the relation between gradient flow and drop in validation loss (performance) for 2-layer and 10-layer GCNs for Cora, Citeseer, and Pubmed. It can be observed that across all datasets, 10-layer GCN has poor gradient flow compared to 2-layer GCN and we observed a negligible drop in validation loss during training.

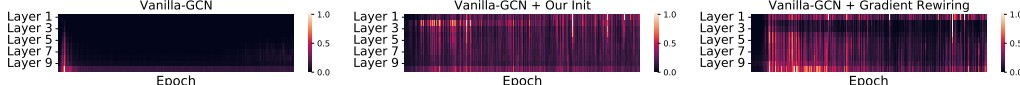

Figure 4: Visualization of the gradient flow of layers during the training of 10-layers vanilla GCN, vanilla-GCN with our new topology-aware initialization and dynamic DE-based rewiring for 500 epochs on Cora. Our methods promote uniform gradient flow across all layers of vanilla-GCNs.

To better understand the behavior of each hidden layer in deep GCN training, we estimated the gradient flow per layer independently of a 10-layer GCN. To our surprise, we found that *many layers receive zero error signals (gradients)* under backpropagation, potentially leading to a catastrophic failure to update during training. Figure 4(a) illustrate layer-wise analysis of gradient flow in 10-layer vanilla-GCN on Cora. We observed that within a few epochs of training, the gradient flow saturates, and then onwards, many layers of GCN receive zero error signal during the backpropagation and they fail to update leading to no drop in validation loss.

| Dataset | Settings | Layer Number | | | | | | | | | | |
|---------|----------|---|---|---|---|---|---|---|---|---|---|---|
| | | 2 | 3 | 4 | 5 | 6 | 7 | 8 | 9 | 10 | 11 | 12 |
| Cora | GCN | 81.1 | 81.9 | 80.4 | 79.9 | 77.5 | 77.1 | 69.5 | 28.2 | 27.4 | 30.1 | 25.4 |
| | GCN + Our Init | 83.0 | 83.4 | 80.6 | 80.0 | 81.2 | 80.6 | 80.4 | 79.9 | 80.1 | 79.2 | 78.5 |
| CiteSeer | GCN | 71.4 | 67.6 | 64.6 | 63.7 | 63.9 | 24.1 | 20.8 | 22.3 | 23.2 | 22.9 | 21.6 |
| | GCN + Our Init | 71.7 | 78.2 | 68.3 | 66.1 | 66.2 | 63.3 | 62.8 | 63.2 | 62.9 | 59.8 | 61.9 |
| PubMed | GCN | 79.0 | 78.9 | 76.5 | 76.7 | 77.1 | 76.5 | 61.2 | 41.8 | 40.7 | 43.2 | 41.0 |
| | GCN + Our Init | 79.3 | 80.0 | 78.5 | 77.6 | 77.6 | 76.7 | 75.8 | 76.9 | 76.2 | 76.3 | 75.9 |

Table 2: Performance Comparison (test accuracy %) of of vanilla-GCNs [1] with and without our newly proposed topology-aware isometric initialization. Experiments are conducted on Cora, Citeseer, and PubMed using vanilla-GCNs with layer $l \in \{2, 3, ..., 12\}$.

Initialization of neural networks is tightly coupled with the propagation of error signals [33] and effect their trainability. Our newly proposed topology-aware isometric initialization and gradient-guided rewiring significantly improve gradient flow (Figure 1(a) and (b)) as well as mitigate the issue of feature collapse of layers in deep vanilla-GCNs. In Figure 4(b) and (c), the detailed layer-wise analysis of vanilla-GCN with our newly proposed methods clearly illustrate how our methods are able to *uniformly restore healthy gradients across all layers* leading to improved trainability and significant performance gains.

| Category | Settings | Cora | | | Citeseer | | | PubMed | | |
|----------|----------|------|------|------|----------|------|------|--------|------|------|
| | | 2 | 16 | 32 | 2 | 16 | 32 | 2 | 16 | 32 |
| Vanilla-GCN | - | 81.10 | 21.49 | 21.22 | 71.46 | 19.59 | 20.29 | 79.76 | 39.14 | 38.77 |
| Skip Connection | Residual | 74.73 | 20.05 | 19.57 | 66.83 | 20.77 | 20.90 | 75.27 | 38.84 | 38.74 |
| | Initial | 79.00 | 78.61 | 78.74 | 70.15 | 68.41 | 68.36 | 77.92 | 77.52 | 78.18 |
| | Jumping | 80.98 | 76.04 | 75.57 | 69.33 | 58.38 | 55.03 | 77.83 | 75.62 | 75.36 |
| | Dense | 77.86 | 69.61 | 67.26 | 66.18 | 49.33 | 41.48 | 72.53 | 69.91 | 62.99 |
| Normalization | BatchNorm | 69.91 | 61.20 | 29.05 | 46.27 | 26.25 | 21.82 | 67.15 | 58.00 | 55.98 |
| | PairNorm | 74.43 | 55.75 | 17.67 | 63.26 | 27.45 | 20.67 | 75.67 | 71.30 | 61.54 |
| | NodeNorm | 79.87 | 21.46 | 21.48 | 68.96 | 18.81 | 19.03 | 78.14 | 40.92 | 40.93 |
| | CombNorm | 80.00 | 55.64 | 21.44 | 68.59 | 18.90 | 18.53 | 78.11 | 40.93 | 40.90 |
| Random Dropping | DropNode | 77.10 | 27.61 | 27.65 | 69.38 | 21.83 | 22.18 | 77.39 | 40.31 | 40.38 |
| | DropEdge | 79.16 | 28.00 | 27.87 | 70.26 | 22.92 | 22.92 | 78.58 | 40.61 | 40.50 |
| | LADIES | 77.12 | 28.07 | 27.54 | 68.87 | 22.52 | 22.60 | 78.31 | 40.07 | 40.11 |
| Identity Mapping | - | **82.98** | 67.23 | 40.57 | 68.25 | 56.39 | 35.28 | 79.09 | 79.55 | 73.74 |
| **Gradient-Guided Rewiring (Ours)** | | 82.79 ±0.16 | **80.19** ±0.32 | **80.01** ±0.15 | **71.06** ±0.21 | **68.54** ±0.11 | **68.49** ±0.31 | **78.90** ±0.20 | **78.32** ±0.19 | **78.51** ±0.27 |

Table 3: Performance Comparison of Gradient-guided Rewiring with respect to various proposed fancy techniques to improve the GCN training. Note that our results are generated using vanilla-GCN [1]. Experiments are conducted on Cora, Citeseer, and PubMed with 2/16/32 layers GCN.

### 3.3 Vanilla-GCNs and our proposed methods

In this section, we conduct a systematic study to understand the performance gain by improving gradient flow using our proposed methods: topology-aware isometric initialization and Gradient Guided Dynamic Rewiring, by incorporating them into the training process of deep vanilla-GCNs. Table 2 demonstrate the performance comparison of vanilla-GCNs [1] with and without our new initialization method with an increase in depth. Our experiments on Cora, Citeseer, and PubMed using vanilla-GCNs with layers $l \in \{2, 3, ..., 12\}$ illustrate how our new initialization has been successful in retaining the trainability of vanilla-GCNs with increasing depth along with improving performance at shallow depths. The highlighted (red) box represents the depths at which vanilla-GCNs lose trainability, i.e, validation loss shows no improvement during training.

Trainability issue of deep GCNs has been extensively studied from the perspective of *over-smoothening* [20, 21] and *information bottleneck* [22] and many approaches broadly categorized as *architectural tweaks* [23, 24, 20, 25], and *regularization & normalization* [26, 27, 28, 29] has been proposed for mitigation. We defer their detailed description to Appendix **??**. Table 3 demonstrates the performance comparison of our gradient-guided dynamic rewiring using skip-connections with respect to various state-of-the-art techniques to improve deep GCNs training. It can be clearly observed that our on-demand dynamic rewiring method of introducing new skip-connections significantly outperforms all fancy normalization and regularization techniques as well as comfortably beats ad-hoc skip-connections techniques.

| Method | Cora | | | | | | PubMed | | | | | |
|---|---|---|---|---|---|---|---|---|---|---|---|---|
| | 2 | 4 | 8 | 16 | 32 | 64 | 2 | 4 | 8 | 16 | 32 | 64 |
| Vanilla-GCN [1] | 81.1 | 80.4 | 69.5 | 21.5 | 21.2 | 21.9 | 79.0 | 76.5 | 61.2 | 39.1 | 38.7 | 35.3 |
| GAT [3] | 81.9 | 80.3 | 31.3 | 30.5 | 27.1 | 27.9 | 78.4 | 77.4 | 29.1 | 26.3 | 28.7 | 25.0 |
| JKNet [50] | 79.1 | 79.2 | 75.0 | 72.9 | 73.2 | 71.5 | 77.8 | 68.7 | 67.7 | 69.8 | 68.2 | 63.4 |
| SGC [52] | 79.3 | 79.0 | 77.2 | 75.9 | 68.5 | 65.3 | 78.0 | 73.1 | 70.9 | 69.8 | 66.6 | 63.2 |
| InceptionGCN [51] | 79.2 | 77.6 | 76.5 | 81.7 | 81.7 | 80.0 | 78.5 | 77.7 | 77.9 | 74.9 | 74.1 | 74.3 |
| GCNII [24] | 82.2 | 82.6 | 84.2 | 84.6 | 85.4 | 85.5 | 78.2 | 78.8 | 79.3 | 80.2 | 79.8 | 79.7 |
| **Ours** | **83.1** | 82.8 | 82.4 | 82.3 | 82.1 | 80.4 | **80.2** | 79.6 | 79.8 | 79.5 | 79.9 | 79.7 |
| (std: $\pm$) | 0.25 | 0.23 | 0.31 | 0.24 | 0.30 | 0.22 | 0.16 | 0.29 | 0.11 | 0.08 | 0.17 | 0.24 |

Table 4: Performance Comparison (test accuracy %) of our proposed method (topology-aware initialization and gradient-guided rewiring) with other previous SOTA frameworks on Cora & Pubmed.

### 3.4 Comparison with state-of-the-art methods

To further validate the effectiveness improving gradient flow in vanilla-GCNs, we perform comparisons with previous state-of-the-art frameworks, including SGC[52], GAT[3], JKNet[50], APPNP[47], InceptionGCN[51], GPRGNN[54], and GCNII[24]. We first observe that Gradient guided rewiring and Topology-aware isometric initialization brings orthogonal benefits and combining them helps vanilla-GCNs to achieve better performance. Table 4 reports the mean classification accuracy and the standard deviation on the test set of Cora and Pubmed of our methods using vanilla-GCNs with layer $l \in \{2, 4, 8, 16, 32, 64\}$. One key benefit to note is that with the help of our methods, we can train vanilla-GCNs as deep as 64 layers. It can be observed that vanilla-GCNs with our methods outperform all SOTA methods on Cora (layers 2 and 4) and Pubmed (layers 2, 4, 8, and 64), while holding the second position for all other layers. To evaluate on a large graph, we have chosen OGBN-AxRiv dataset, and Table 5 represents the performance comparison of vanilla-GCNs with our method against SOTA baselines. It can be observed that with the help of our methods, vanilla-GCNs can be efficiently trained deeper without any significant loss in performance. Very similar to Cora and Pubmed, our methods beats all SOTA methods while training 2-layer vanilla-GCNs and holds second position for all other layers. It is interesting to note that for OGBN-AxRiv dataset, vanilla-GCNs with our proposed method consistently have performance improvement with an increase in depth possibly because of being able to capture long-term information effectively, which is in contrast to most of the SOTA methods whose performance suffers with increased in depth. Note that we are NOT trying to beat SOTA, but instead, our objective is to reveal the correct initialization and training scheme for vanilla-GCNs, and scale deep efficiently.

The last sanity check is whether our proposed methods can make deep vanilla-GCNs effective across multiple different graph datasets. Specially, we evaluate it on seven other open-source graph datasets: (i) one Co-author datasets [55] (CS), (ii) two Amazon datasets [55] (Computers and Photo), (iii) three

| Layer | GCN[1] | SGC[52] | DAGNN [25] | GCNII[24] | JKNet[50] | APPNP[47] | GPRGNN[54] | **Ours** |
|---|---|---|---|---|---|---|---|---|
| 2 | 69.46±0.22 | 61.98±0.08 | 67.65±0.52 | **71.24±0.17** | 63.73±0.38 | 65.31±0.23 | 69.31±0.09 | **71.59±0.08** |
| 16 | 67.96±0.38 | 41.58±0.27 | **71.82±0.28** | 72.61±0.29 | 66.41±0.56 | 66.95±0.24 | 70.30±0.15 | 71.76±0.03 |
| 32 | 45.46±4.50 | 34.22±0.04 | 71.46±0.27 | **72.60±0.25** | 66.31±0.63 | 66.94±0.26 | 70.18±0.16 | **72.03±0.55** |
| 64 | 38.40±7.63 | 36.17±2.11 | 70.22±1.54 | **72.13±0.79** | 62.97±3.81 | 65.54±1.74 | 70.59±2.60 | **72.28±0.92** |

Table 5: Performance Comparison (test accuracy %) of our proposed method (topology-aware initialization and Gradient-guided rewiring) with previous SOTA frameworks using OGBN-AxRiv.

WebKB datasets [56] (Texas, Wisconsin, Cornell), and (iv) the Actor dataset [56]. Table 6 reports the performance comparison of deep vanilla-GCN with 32-layers with state-of-the-art methods having the same depth. Our methods universally encourage significant trainability benefits to deep vanilla-GCNs across all datasets achieving state-of-the-art performance on Computers dataset while performing in top-2 for all remaining datasets.

| Category | CS [55] | Computers [55] | Photo [55] | Texas [56] | Winconsin [56] | Cornell [56] | Actors [56] |
|---|---|---|---|---|---|---|---|
| GCN [1] | 24.01±3.42 | 58.72±4.97 | 58.64±8.40 | 60.12±4.22 | 52.94±3.99 | 54.05±7.11 | 25.46±1.43 |
| GAT [3] | 11.04±0.94 | 9.42±0.54 | 17.11±1.02 | 11.54±0.72 | 14.01±0.88 | 19.78±1.42 | 6.42±0.39 |
| SGC [52] | 70.52±3.96 | 37.53±0.20 | 26.60±4.64 | 56.41±4.25 | 51.29±6.44 | 58.57±3.44 | 26.17±1.15 |
| GCNII [24] | 71.67±2.68 | 37.56±0.43 | 62.95±9.41 | **69.19±6.56** | **70.31±4.75** | **74.16±6.48** | **34.28±1.12** |
| JKNet [50] | 81.82±3.32 | **67.99±5.07** | **78.42±6.95** | 61.08±6.23 | 52.76±5.69 | 57.30±4.95 | 28.80±0.97 |
| APPNP [47] | **91.61±0.49** | 43.02±10.16 | 59.62±23.27 | 60.68±4.50 | 54.24±5.94 | 58.43±3.74 | 28.65±1.28 |
| **Ours** | 89.33±2.10 | **77.18±1.72** | 72.77±2.27 | 64.28±2.93 | 59.19±9.07 | 58.51±1.66 | 30.95±1.04 |

Table 6: Transfer studies of our proposed method (topology-aware initialization and gradient-guided rewiring) with deep vanilla-GCNs (32-layers). Comparisons are conducted on seven open-source widely adopted datasets with other previous state-of-the-art frameworks.

## 3.5 Dirichlet Energy-based analysis of our proposed methods

In this section, we use Dirichlet Energy to illustrate how our proposed techniques help in mitigating the issue of losing expressiveness, enable deep GNNs to leverage the high-order neighbors. Node pair distance has been widely adopted to quantify the embedding similarities, and Dirichlet energy is simple and expressive metric to estimate the expressiveness of node embeddings learned by GCNs, in a topology-aware fashion [46, 57]. Following [46], we define Dirichlet Energy as:

**Definition 2.** [46] Dirichlet energy $E(\boldsymbol{x})$ of a scalar function $\boldsymbol{x} \in \mathbb{R}^N$ on the graph G is defined as:

$$E(\boldsymbol{x}) = \boldsymbol{x}^T \boldsymbol{L} \boldsymbol{x} = \frac{1}{2} \sum_{(i,j)\in\mathcal{E}} \left( \frac{x_i}{(1+d_i)^{1/2}} - \frac{x_j}{(1+d_j)^{1/2}} \right)^2 \tag{17}$$

For the vector field $\boldsymbol{X} = [\boldsymbol{x}_1 \quad \cdots \quad \boldsymbol{x}_N] \in \mathbb{R}^{C \times N}$, where $\boldsymbol{x}_i \in \mathbb{R}^C$, Dirichlet energy is defined as:

$$E(\boldsymbol{X}) = \text{Tr}(\boldsymbol{X}\boldsymbol{L}\boldsymbol{X}^T) = \frac{1}{2} \sum_{(i,j)\in\mathcal{E}} \left\| \frac{\boldsymbol{x}_i}{(1+d_i)^{1/2}} - \frac{\boldsymbol{x}_j}{(1+d_j)^{1/2}} \right\|_2^2 \tag{18}$$

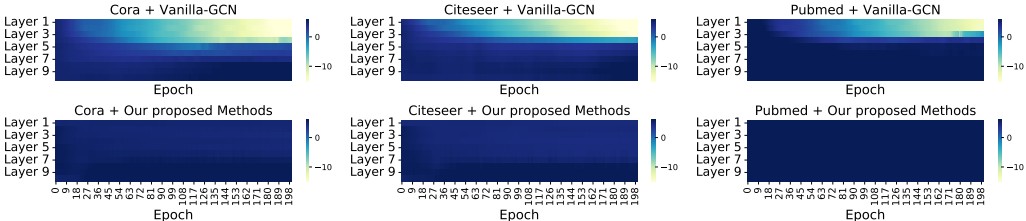

Figure 5: Visualization of Dirichlet Energy of layers during the training of 10-layers vanilla GCN (row 1) and with our proposed methods on Cora, Pubmed, and Citeseer. Plotted using the log scale for better visualization. Our methods prevent Dirichlet Energy to drop to zero during training and maintain the expressiveness of feature embeddings.

We track the Dirichlet energy of feature matrix w.r.t. the (augmented) normalized Laplacian at different layers of the vanilla-GCN during training and observed that during training of a deep vanilla-GCN, the Dirichlet energy of some layers decreases exponentially and becomes close to zero within a few epochs of training, i.e lose expressiveness (Figure 5 row 1). Our newly proposed methods: topology-aware isometric initialization and gradient-guided rewiring significantly improve gradient flow, as well as mitigate the issue of exponentially dropping Dirichlet energy across all the layers of deep GCNs, thereby restoring the expressiveness of node embeddings (Figure 5 row 2).

## 4 Other Related Works

GNNs have established state-of-the-art performance in numerous real-world applications [11, 12, 13, 11, 5, 4, 14, 15, 16, 12, 17, 18]. While deep architectures improve the representational power of neural networks [58, 59], not every useful GNN has to be deep. Many real-world graph are "small-world"[60], where a node can reach any other node in a few hops, hence a few layers would suffice to provide global coverage. However, when the graph data has no small-world property, or the related task requires long-range information, then deeper GNNs become very necessary. Many works [20, 21] revealed that when we start stacking spatial aggregations recursively in GNNs, the node representations will collapse to indistinguishable vectors and it will hamper the training of deep GNNs. More generally, this phenomenon happens for any message-passing mechanism via stochastic matrices including attention [61]. Recently, there has been a series of techniques developed to handle the over-smoothing issue, which can be broadly categorized under skip connection, graph normalization, random dropping. Skip-connections [23, 24, 38, 20] have been applied to GNNs to exploit node embeddings from the preceding layers, to relieve the over-smoothing issue. Graph normalization techniques [28, 62, 63, 64, 65] re-scale node embeddings over an input graph to constrain pairwise node distance and thus alleviate over-smoothing. Dropout methods [66, 26, 27, 67] can be regarded as data augmentations, which help relieve both the over-fitting and over-smoothing issues in training very deep GNNs.

## 5 Conclusion

This paper makes an important step towards understanding the substandard performance of deep vanilla-GCNs from the signal propagation perspective and hypothesizes that by facilitating healthy gradient flow, we can significantly improve their trainability, as well as achieve state-of-the-art (SOTA) level performance or close using merely vanilla-GCNs. This paper derives a topology-aware isometric initialization scheme for vanilla-GCNs based on the principles of isometry. Additionally, this paper proposes to use Gradient Flow for the dynamic rewiring of vanilla-GCNs with skip-connections. Extensive experiments across multiple datasets illustrate that these methods improvise gradient flow in deep vanilla-GCNs and significantly boost their performance. An interesting direction for future work includes building theoretical relations between our proposed methods and gradient flow.

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
