# A Derivation of Eq. 12 and 13

Recall that, in Section 2.3 our goal is to make initialized GCN isometric. Instead of exactly compute such initialization by solving an optimization, we propose the following objective, which has a closed-form solution:

$$\boldsymbol{W}^* = \underset{\boldsymbol{W} \in \mathbb{R}^{C' \times C}}{\operatorname{argmin}} \sum_{i,j \in [N]} \left\| \left(\boldsymbol{a}_i \boldsymbol{a}_j^\top\right) \otimes \left(\boldsymbol{W}^\top \boldsymbol{W}\right) - \left(\boldsymbol{e}_i \boldsymbol{e}_j^\top\right) \otimes \boldsymbol{I}_C \right\|_F^2. \tag{19}$$

In this section, we give the detailed derivation to reach the closed-form solution. Notice that both $\boldsymbol{a}_i \boldsymbol{a}_j^\top$ and $\boldsymbol{e}_i \boldsymbol{e}_j^\top$ are sparsely structured. Let $\boldsymbol{A}^{(i,j)} = \boldsymbol{a}_i \boldsymbol{a}_j^\top$ and $\boldsymbol{E}^{(i,j)} = \boldsymbol{e}_i \boldsymbol{e}_j^\top$, then we have:

$$\boldsymbol{A}_{m,n}^{(i,j)} = \begin{cases} 1 & (i,m) \in \mathcal{E} \text{ and } (j,n) \in \mathcal{E} \\ 0 & \text{Otherwise} \end{cases}, \qquad \boldsymbol{E}_{m,n}^{(i,j)} = \begin{cases} 1 & m = i \text{ and } n = j \\ 0 & \text{Otherwise} \end{cases}, \tag{20}$$

where $\boldsymbol{A}_{m,n}^{(i,j)}, \boldsymbol{E}_{m,n}^{(i,j)}$ denotes the $m$-th row and the $n$-th column entry of $\boldsymbol{A}^{(i,j)}, \boldsymbol{E}^{(i,j)}$, respectively. Hence, for $m, n \in [N]$ such that $(i,m) \in \mathcal{E}$ and $(j,n) \in \mathcal{E}$ but $m \neq i$ or $n \neq j$, the block-wise difference at location $(m,n)$ remains $\boldsymbol{W}^\top \boldsymbol{W}$. While for $m = i$ and $n = j$, the block difference is $\boldsymbol{W}^\top \boldsymbol{W} - \boldsymbol{I}_C$ as $\boldsymbol{A}$ contains self-loop (i.e., $(i,i) \in \mathcal{E}$ and $(j,j) \in \mathcal{E}$). Then we can simply our objective Eq. 19 as below:

$$\boldsymbol{W}^* = \underset{\boldsymbol{W} \in \mathbb{R}^{C' \times C}}{\operatorname{argmin}} \sum_{i,j \in [N]} \left( \sum_{\substack{m,n \in [N] \\ (m,n) \neq (i,j)}} \left\| \boldsymbol{A}_{m,n}^{(i,j)} \boldsymbol{W}^\top \boldsymbol{W} \right\|_F^2 + \left\| \boldsymbol{W}^\top \boldsymbol{W} - \boldsymbol{I}_C \right\|_F^2 \right) \tag{21}$$

$$= \underset{\boldsymbol{W} \in \mathbb{R}^{C' \times C}}{\operatorname{argmin}} \sum_{i,j \in [N]} (d_i + d_j + d_i d_j) \left\| \boldsymbol{W}^\top \boldsymbol{W} \right\|_F^2 + N^2 \left\| \boldsymbol{W}^\top \boldsymbol{W} - \boldsymbol{I}_C \right\|_F^2. \tag{22}$$

To solve Eq. 22, we decompose the diagonal and off-diagonal components from $\boldsymbol{W}^\top \boldsymbol{W}$. For off-diagonal terms, we can minimize them to zeros, thus we have $\boldsymbol{w}_k^\top \boldsymbol{w}_l = 0$, where we note that $\boldsymbol{w}_k$ denotes the $k$-th column of $\boldsymbol{W}$. For on-diagonal terms, it is equivalent to minimizing $\sum_{i,j \in [N]} (d_i + d_j + d_i d_j) \|\boldsymbol{w}_k\|_2^4 + N^2 (\|\boldsymbol{w}_k\|_2^2 - 1)^2$ for every $k \in [C]$. Then combining both arguments above, the optimal solution should satisfy:

$$\|\boldsymbol{w}_k\|_2^2 = \frac{N^2}{\sum_{i,j \in [N]} (d_i + 1)(d_j + 1)} \qquad \forall k \in [C], \tag{23}$$

$$\boldsymbol{w}_k^\top \boldsymbol{w}_l = 0 \qquad \forall k, l \in [C], k \neq l, \tag{24}$$

which reaches our final conclusion in Eq. 14. To compute this magnitude given a graph, one needs to first compute the degree of each node, and plug them into our Eq. 12. The complexity to compute this value is as small as $\mathcal{O}(N^2)$, which is significantly lower than directly optimizing Eq. 10.

# B Description of Methods in Table 3

Table 3 presents a comparison of our Gradient-Guided Rewiring dynamic rewiring with respect to various fancy methods to improve the trainability of deep vanilla-GCNs [1] such as skip-connections, regularization. Our formal description of these methods are:

## B.1 Skip Connections

Skip-connections [23, 24, 38, 20] helps in alleviating the problem of over-smoothing and significantly improve the accuracy and training stability of deep GCNs. For a $L$ layer GCN, we can apply various type of skip-connections after certain graph convolutional layers with the current and preceding embeddings $X^l, 0 \leq l \leq L$. We have compared our method with the following four representative types of skip connections:

1. *Residual Connection:* $\boldsymbol{X}^l = (1 - \alpha) \cdot \boldsymbol{X}^l + \alpha \cdot \boldsymbol{X}^{l-1}$

2. *Initial Connection:* $\boldsymbol{X}^l = (1 - \alpha) \cdot \boldsymbol{X}^l + \alpha \cdot \boldsymbol{X}^0$

3. *Dense Connection:* $\boldsymbol{X}^l = \text{COM}(\{\boldsymbol{X}^k, 0 \le k \le l\})$

4. *Jumping Connection:* $\boldsymbol{X}^L = \text{COM}(\{\boldsymbol{X}^k, 0 \le k \le L\})$

where $\alpha$ is *residual* and *initial connections* is a hyperparameter to weight the contribution of a node features from the current layer $l$ and previous layers. *Jumping connection* is a simplified case of *dense connection* and it is only applied at the end of the whole forward propagation process to combine the node features from all previous layers.

## B.2 Graph Normalization

Graph normalization [28, 62, 63, 64, 65] re-scale node embeddings over an input graph to constrain pairwise node distance and thus alleviate over-smoothing. Our investigated normalization mechanisms are formally depicted as follows:

1. *BatchNorm:* $\boldsymbol{x}_{:,j} = \gamma \cdot \frac{\boldsymbol{x}_{:,j} - \mathbb{E}(\boldsymbol{x}_{:,j})}{\text{std}(\boldsymbol{x}_{:,j})} + \beta$

2. *PairNorm:* $\widetilde{\boldsymbol{x}}_i = \boldsymbol{x}_i - \frac{1}{n} \sum_{i=1}^{n} \boldsymbol{x}_i$, $\text{PairNorm}(\boldsymbol{x}_i; s) = \frac{s \cdot \widetilde{\boldsymbol{x}}_i}{(\frac{1}{n} \sum_{i=1}^{n} \|\widetilde{\boldsymbol{x}}_i\|_2^2)^{1/2}}$

3. *NodeNorm:* $\text{NodeNorm}(\boldsymbol{x}_i; \ p) = \frac{\boldsymbol{x}_i}{\text{std}(\boldsymbol{x}_i)^{1/p}}$

where $\boldsymbol{x}_{:,j} \in \mathbb{R}^N$ denotes the $j$-th row of $\boldsymbol{X}$ (the $j$-th channel), $\boldsymbol{x}_i \in \mathbb{R}^C$ denotes the $i$-th column of $\boldsymbol{X}$ (the $i$-th node features), $s$ in *PairNorm* is a hperparameter controlling the average pair-wise variance and $p$ in *NodeNorm* denotes the normalization order.

# C Dataset Details

Table 7 provided provides the detailed properties and download links for all adopted datasets. We adopt the following benchmark datasets since i) they are widely applied to develop and evaluate GNN models, especially for deep GNNs studied in this paper; ii) they contain diverse graphs from small-scale to large-scale or from homogeneous to heterogeneous; iii) they are collected from different applications including citation network, social network, etc.

| Dataset | Nodes | Edges | Features | Classes | Download Links |
|---------|-------|-------|----------|---------|----------------|
| Cora | 2,708 | 5,429 | 1,433 | 7 | https://github.com/kimiyoung/planetoid/raw/master/data |
| Citeseer | 3,327 | 4,732 | 3,703 | 6 | https://github.com/kimiyoung/planetoid/raw/master/data |
| PubMed | 19,717 | 44,338 | 500 | 3 | https://github.com/kimiyoung/planetoid/raw/master/data |
| OGBN-ArXiv | 169,343 | 1,166,243 | 128 | 40 | https://ogb.stanford.edu/ |
| CoauthorCS | 18,333 | 81,894 | 6805 | 15 | https://github.com/shchur/gnn-benchmark/raw/master/data/npz/ |
| Computers | 13,381 | 245,778 | 767 | 10 | https://github.com/shchur/gnn-benchmark/raw/master/data/npz/ |
| Photo | 7,487 | 119,043 | 745 | 8 | https://github.com/shchur/gnn-benchmark/raw/master/data/npz/ |
| Texas | 183 | 309 | 1,703 | 5 | https://raw.githubusercontent.com/graphdml-uiuc-jlu/geom-gcn/master |
| Wisconsin | 183 | 499 | 1,703 | 5 | https://raw.githubusercontent.com/graphdml-uiuc-jlu/geom-gcn/master |
| Cornell | 183 | 295 | 1,703 | 5 | https://raw.githubusercontent.com/graphdml-uiuc-jlu/geom-gcn/master |
| Actor | 7,600 | 33,544 | 931 | 5 | https://raw.githubusercontent.com/graphdml-uiuc-jlu/geom-gcn/master |

Table 7: Graph datasets statistics and download links.