# OpenReview forum: "Old can be Gold: Better Gradient Flow can Make Vanilla-GCNs Great Again"
_NeurIPS.cc/2022/Conference — NeurIPS 2022 Accept_

### Official Review · Reviewer_K9SU · 2022-07-09

**Rating:** 8
**Confidence:** 4
**Soundness:** 4 excellent
**Presentation:** 4 excellent
**Contribution:** 4 excellent

**Summary:**

This work adds insight into why Graph Convolutional Networks (GCNs) have had trouble going deep. In particularly, it extends the intuition of "over-smoothening" and "information bottlenecks" that plague other neural nets (which were solved by gradient flow) and uses techniques to improve gradient flow for GCNs. In particular, it does better initialization (topology-aware isometric initialization) and rewiring (Dirichlet Energy Guided Dynamic Rewiring). By having a better distribution of initial weights and adding skip connections to propagate gradients into needed areas on where needed, the authors demonstrate very competitive (usually top-2 of 14) performance by enhancing a mere vanilla-GCN.

**Questions:**

Under what conditions is the sparsity assumption for the structure of a_{i}a_{j} no longer true?

**Strengths And Weaknesses:**

Strengths
+ This is a very well-motivated work. The discussion of gradient flow solutions for other neural networks very cleanly tees up the "obviousness" of gradient flow solutions for GCNs.
+ The experiments to measure gradient flow in GCNs (eg Figure 2) are useful and interesting.
+ Really enjoyed the simple insight applied by this paper to demonstrate big gains. Particularly impressed by the dynamic skip connections.
+ The writing for this paper was very clear (e.g. description of how GCNs work in section 2.2, underlining certain phrases for emphasis, approachable tone of describing other methods as "fancy state-of-the-art methds")

Weaknesses
- It wasn't always clear whether the demonstrated issues with deep networks were primarily from gradient flow, per se, as opposed to generally just overfitting. Deeper networks are higher-capacity, but the motivating experiments did not disentangle that.
- You reference "Appendix 1" on line 158, but this work does not have an appendix.

Suggested Tweaks
- Format-wise, the figures were a little scattered. Figure 1 is listed on page 2 but not described until page 4.
- There are a few typos (e.g. "bolltleneck" on line 32, plural complex numbers* on line 138) or seemingly wrong words (eg on page 5 you miight have meant "improvises" instead of "improves" gradient flow, though I don't think the next word is correct when you say "Our dynamic rewiring ... retain Dirichlet Energy").
- In Table 4, imo you should only bold the best entry in each column. If you really want to draw attention to the second best entry, perhaps you could underline them?

---

> ### Author Response · Authors · 2022-08-01
> **Author Response for Reviewer K9SU**
>
> Many thanks for identifying the significance of our work we appreciate your recommendation as a strong accept. We are glad that you found our work well-motivated and enjoyable to read. We would like to address some concerns pointed by you point-by-point below:
>
> **[Cons1] Is trainability issue of deep GNNs is due to overfitting?** We would like to highlight that the trainability of deep GCNs are not primarily due to overfitting, as during the training when deep GNNs stack spatial aggregations recursively, the node representations collapse to indistinguishable vectors and hurts training further. Low trainability instead leads to underfitting. As the training progress for deep GCNs, we observed that even the **training loss doesn't reduce and network performance on the training data is significantly low (underfitting)**.  A very recent benchmarking paper Chen et. al. TPAMI 2022 [1], illustrated that even with large datasets like OGBN-ArXiv, the performance of deep GCNs significantly drops from ~70% $\rightarrow$ ~45% when we increase layer count from 2$\rightarrow$32. After applying our techniques, we not only reduce the generalization gap, but also lower the training error simultaneously.
>
> **[Cons2] Missing Appendix?** Thank you for pointing this out. Our appendix was uploaded with the supplementary section of our submission.  We have added it in the main revised draft.
>
> **[Cons3] Under what conditions is the spasrity assumption for the structure of a_{i}a_{j} no longer hold true?**
> $a_{i}$ $a_{j}$ denotes the i-th and j-th column of (A+I). Therefore, $a_{i} a_{j}^\top$ has $(1+d_i)(1+d_j)$ non-zero entries. In general,  $a_{i} a_{j}^\top$ is presumably very sparse, but if the degree of i,j is very high, $a_{i} a_{j}^\top$ can be dense. Nevertheless, our derivation in the supplementary material does not rely on the assumption of its sparsity.
>
> **[Cons4] Suggested Tweaks?** We are again very thankful for your valuable suggestions to improve typos, figures, table entries and we have updated our revised draft with your suggestions.
>
>
> [1] Chen, T., Zhou, K., Duan, K., Zheng, W., Wang, P., Hu, X., & Wang, Z. (2022). Bag of Tricks for Training Deeper Graph Neural Networks: A Comprehensive Benchmark Study. IEEE transactions on pattern analysis and machine intelligence, PP.

---

### Official Review · Reviewer_7SM5 · 2022-07-12

**Rating:** 4
**Confidence:** 2
**Soundness:** 3 good
**Presentation:** 3 good
**Contribution:** 2 fair

**Summary:**

The work focuses on improving the trainability of vanilla-GCN. Starting from the initial observation of the gradient flow issue, the authors propose two techniques, namely topology-aware isometric initialization and Dirichlet energy guided rewiring to improve the optimization. Empirically, on several benchmark datasets, authors show that the propose two techniques do improve the training of vanilla-GCN, leading to a competitive model quality (though not SOTA).

**Questions:**

Can the proposed two techniques improve SOTA GNNs not just GCNs?


**Limitations:**

Nothing particular.

**Strengths And Weaknesses:**

The paper is written in a very logical way and the proposed methods are well motivated. The empirical evaluation is systematic and convincing.

However, it seems the trainability issue generally happens when the number of layers is large. But even when the proposed methods are used, increasing the number of layers generally does not improve the model quality at all. Hence, is it possible that the trainability issue comes from the fact that the datasets are too small and/or too noisy such that higher model capacity does not help?

Another question I have is that would the proposed two techniques be helpful to other more complicated SOTA GNNs with e.g. attention  & normalization? If so, the proposed techniques would be more helpful in practice.

Finally, while adaptive rewiring based on Dirichlet energy is quite effective, it always uses the output feature matrix of first layer as the skip feature. Compared to the standard skip connection used in ResNet, this effectively makes the network relatively shallower, which can be seen as trading model complexity for trainability. Then, why not just use shallower networks instead then?

---

> ### Author Response · Authors · 2022-08-01
> **Author Response for Reviewer 7SM5**
>
> We are very thankful for your time to review our work and appreciate that you found our work very well motivated and empirically convincing and systematic. We would like to address your concerns point by point:
>
> **[Cons1] Is it possible that the trainability issue comes from the fact that datasets are too small/noisy that model capacity doesn’t help?** We would like to highlight that the trainability issue of deep GNNs is a well-studied problem and it is not merely because of the underlying dataset. As deep GNNs stack spatial aggregations recursively, the node representations will collapse to indistinguishable vectors. Such over-smoothing phenomenon hinders the training of deep GNNs and the dependency modeling to high-order neighbor information. A very recent benchmarking paper Chen et. al. TPAMI 2022, illustrated that **even with large datasets like OGBN-ArXiv, the performance of deep GCNs significantly drops from ~70% $\rightarrow$ ~45% when we increase layer count from 2$\rightarrow$32**. Our work provides an interesting approach to understanding the issue of trainability from the gradient propagation perspective and suggests simple and effective approaches to train very deep GNNs without any loss in performance.
>
> **[Cons2] Proposed techniques help in improving complicated GNNs with attention and normalization?** Note that the first technique isometric initialization is a generalized idea and it has been successfully adapted to improve gradient flow in many other architectures such as CNNs (Qi et. al. ICML 2020 [2,3]) whereby enforcing the convolution kernels to be near isometric during initialization, the authors have been able to effectively train deep vanilla ConvNets without normalization nor skip connections can also be trained to achieve surprisingly good performance on standard image recognition benchmarks. Our proposed initialization can be derived for complex backbones such as GAT but the scope of this work is limited to vanilla-GCNs and showing how by incorporating our techniques to facilitate healthy gradient flow in deep GCNs, their performance can be significantly improved. Surprisingly, some recent experiments by simply adapting our initialization to SOTA normalization methods illustrate performance benefits (eg. with PairNorm we 16 layer and 32 layer GCN, we observed around ~12.5% and ~33.9% improvement for Cora), which is highly encouraging. Similarly, our dynamic rewriting strategy with PairNorm also benefits deep GCNs, where for a 32-layer GCN on Cora we observed a performance increase from 17.27% $\rightarrow$ 71.56%, which clearly illustrates the adaptability of our techniques.
>
> **[Cons3] Why not use the shallower network instead?** Note that our adaptive rewiring strategy is defined as $X^{l} =\mathbb{1}{[E(X^{(l_t)}) < pE(X^{(l_0)})]} \alpha * X^{(1)} + \tilde{X}^{l}$ where $\mathbb{1}[\cdot]$ is an indicator function, $t$ denotes the training epoch, $p$ denotes energy drop threshold, and $\tilde{X}^{l} = W_{l-1}X^{(l-1)}(A+I)$. We would like to highlight that our rewiring strategy is decided at the training time when we observe that the DE energy of a layer drops below p% of its initial value. It is quite possible that many intermediate layers of a deep require no rewiring, and medium depth GCNs requires no rewiring at all, and our method allows this flexible architecture adaptation. With increasingly large graph datasets with millions of nodes and edges, we need a more expressive network with large parameter counts to avoid underfitting. Recently a huge effort has been towards improving the trainability of deep GCNs, because stacking more layers to improve the expressive power of GNNs and the ability to extract information from high-order neighbors, leads to the issue of over-smoothening where the features of the nodes in the (connected) graph would converge to similar values. With the usage of dynamic residual connections, we are not essentially trading model complexity for trainability, but instead ensuring that the final representation of each node retains at least a fraction of $\alpha$ from the initial layer even if we stack many layers to extract information from high-order neighbors.
>
> We hope our responses have clarified many of your concerns, and please do not hesitate to let us know what else we could do in order to convince you for a rating upgrade.
>
> [1] Chen, T., Zhou, K., Duan, K., Zheng, W., Wang, P., Hu, X., & Wang, Z. (2022). Bag of Tricks for Training Deeper Graph Neural Networks: A Comprehensive Benchmark Study. IEEE transactions on pattern analysis and machine intelligence, PP.
>
> [2] Qi, Haozhi, et al. "Deep isometric learning for visual recognition." International Conference on Machine Learning. PMLR, 2020.
>
> [3] Lee, N., Ajanthan, T., Gould, S., & Torr, P.H. (2020). A Signal Propagation Perspective for Pruning Neural Networks at Initialization. ICLR 2020

---

> ### Author Response · Authors · 2022-08-05
> **Author Response for Reviewer 7SM5**
>
> Dear Reviewer 7SM5,
>
> We thank you for your time to review our work and your constructive comments to improve it, and we really hope to have a further discussion with you to see if our response solves your concerns. We have replied to the important points raised by you such as if the trainability issue of deep GCNs comes from small/noisy datasets, the applicability of our techniques for GNNs with normalization and attention, etc in our rebuttal response. Since the author-reviewer discussion period has started for a few days, we will appreciate if you could check our response to your review comments soon. This way, if you have further questions and comments, we can still reply before the author-reviewer discussion period ends.
>
> If our response resolves your concerns, we kindly ask you to consider raising the rating of our work. We again thank you for your time and efforts.
>
> Best, Authors of Paper7865

---

> ### Author Response · Authors · 2022-08-08
> **Author Response for Reviewer 7SM5**
>
> Dear Reviewer 7SM5,
>
> Since the author-reviewer discussion period is about to get over in the next ~24 hours, we are very eagerly waiting for your response to our rebuttal comments. We sincerely hope that you have read our rebuttal comments, and if you have any further doubts, please let us know as soon as possible so that we can respond to them within the time limits. If our response resolves your concerns, we kindly ask you to consider raising the rating of our work.
>
> Thanks! Authors

---

### Official Review · Reviewer_ePpt · 2022-07-12

**Rating:** 5
**Confidence:** 5
**Soundness:** 3 good
**Presentation:** 3 good
**Contribution:** 3 good

**Summary:**

This paper tries to improve the performance of the deep GNN model by improving the gradient flow during training. Specifically, the authors propose two methods, topology-
aware isometric initialization and dynamic rewiring by Dirichlet Energy. Topology-
aware isometric initialization aims to provide a good initialization with isometric property, while dynamic rewiring is to dynamically add the skip connections based on the decrease of  Dirichlet energy of each GCN layer.


**Questions:**

Please refer  to the Strengths And Weaknesses.

**Ethics Review Area:**

["I don’t know"]

**Strengths And Weaknesses:**

Strengths:
The idea of improving gradient flow is insightful. I agree that gradient flow is a key factor in improving the performance of Deep GNNs.
It's easy to read and understand. I enjoyed reading this paper.

Weaknesses:
1. I think the observations in this paper are insightful. The proposed solution generally works.  The main weakness lies in the theoretical analysis. In the isometric initialization part, the authors only give the construction of the initialization (Eq 14) without any further justification. Then, in the rewiring part, the authors directly apply the Dirichlet energy as a measure to control the generation of skip connections. It makes use of the property of Dirichlet energy but is too artificial.  Meanwhile, the relationship between Dirichlet energy and gradient flow is not well elaborated.

2. I like the nice figures and rich empirical evaluations in this paper.  Here are my suggestions:

- Since the gradient flow is defined on gradients rather than graphs, I’m curious whether the proposed isometric initialization and dynamic rewiring can improve the gradient flow on the MLP model or not.

- As stated in the paper, the dynamic rewiring needs to update Dirichlet energy $Tr(XLX^{T})$ for each GCN layer during training. It looks like the calculation of  $Tr(XLX^{T})$ is expensive, especially for a large graph. I want to see the impact of this dynamic rewiring in terms of training time.

 - Is isometric initialization also work for other GNN backbones, such as GAT, and APPNP?

---

> ### Author Response · Authors · 2022-08-01
> **Author Response for Reviewer ePpt**
>
> We are very thankful that you found our paper enjoyable to read and understand. We would like to address your concerns point-by-point below:
>
> **[Cons1] Relationship between Dirichlet Energy and gradient flow is not well established?** As argued by Cai et. al. (ICML2020)[1], Dirichlet energy is a conceptually clean and cheap way to measure the expressiveness of feature embeddings. They additionally argued (+proved) that the Dirichlet energy of many GCN layer embeddings will converge to zero, resulting in the loss of discriminative power, with an increase in depth. In our experiments, by closely analyzing the gradients of these layers across multiple datasets and multiple GCN depths, we found that gradients received by these layers are almost zero which prevents any effective training and weight changes, leading to poor performance, and loss of expressiveness. Since DE is a **well-defined, cheap, and architecture-agnostic metric**, based on our strong empirical correlation, in our work, we use it as a proxy to locate layers that suffer from poor gradient flow. Note that gradient flow (magnitude) measurement heavily depends on dataset, architecture, as well as stage of training which demands many hyperparameter thresholds, which is simplified by our cheap proxy of DE where we simply supplement a layer with our dynamic skip as soon as its energy drops to p% of its original value.
>
> **[Cons2] Does our proposed methods can improve MLP models or not?** We would like to highlight that both our isometric initialization and dynamic rewiring are generalized ideas and they can be easily adapted to any neural architecture to improve the gradient flow and trainability. For example, Qi et. al. ICML 2020 [2], have shown by enforcing the convolution kernels to be near isometric during initialization, improves gradient flow, and deep vanilla ConvNets without normalization nor skip connections can also be trained to achieve surprisingly good performance on standard image recognition benchmarks. Lee et al. ICLR 2020 [3], have also illustrated the benefits of isometric initialization for feed-forward MLP networks. Furthermore, Jaiswal et. al ICML 2022, recently used a variant of our dynamic skips (called ghost skips) to improve the gradient flow in sparse neural networks which suffer from poor gradient flow, thereby achieving SOTA performance across multiple architectures and datasets. Similar benefits have also been outlined in Teressa et. al. 2021. Note that there has been sufficient work in the context of initialization to understand the dynamics of learning of neural networks (CNNs/RNNs/MLPs) but for GNNs, we are still lagging and our work is an important direction to reconsider signal propagation perspective to understand the trainability and performance issue of deep GCNs.
>
> **[Cons3] Calculation of Dirichlet energy is expensive?** We would like to highlight that DE can be efficiently implemented as a message passing paradigm during the tracking stage. Considering the real-world graphs are highly sparse, each node only gets its information from its immediate neighbor, the asymptotic cost is approximately O(1) and for all the nodes in the graph is O(N), i.e. scales only linearly with the number of nodes in the graph, which is similar to training a GCN layer. Moreover, note that we track DE only for the first few epochs of training (approximately 100 epochs) and when the troublesome layers are identified and handled using rewiring, the training proceeds as normal without estimating DE. Practically, on RTX A6000 GPUs, we observed marginal (~10ms - 90ms) delay per epoch for the datasets reported in our experiments.
>
> [1] Cai, Chen, and Yusu Wang. "A note on over-smoothing for graph neural networks." arXiv preprint arXiv:2006.13318 (2020).
>
> [2] Qi, Haozhi, et al. "Deep isometric learning for visual recognition." International Conference on Machine Learning. PMLR, 2020.
>
> [3] Lee, N., Ajanthan, T., Gould, S., & Torr, P.H. (2020). A Signal Propagation Perspective for Pruning Neural Networks at Initialization. ICLR 2020
>
> [4] Jaiswal, Ajay Kumar, et al. "Training Your Sparse Neural Network Better with Any Mask." International Conference on Machine Learning. PMLR, 2022.

---

> > ### Author Response · Authors · 2022-08-01
> > **Author Response for Reviewer ePpt**
> >
> > **[Cons4] Is isometric initialization also works for other GCN backbones like GAT, and APPNP?** Note that isometric initialization is a generalized idea and it has been successfully adapted to improve gradient flow in many other architectures such as CNNs (Qi et. al. ICML 2020, Lee et. al ICML 2020)[2,3] and it can be derived for backbones like GAT and APPNP. However, the scope of this work is limited to vanilla-GCNs and showing their competitiveness with SOTA with our isometric initialization. We plan to extend and derive isometric initialization bounds for more complex architecture designs such as attention (GAT) and APPNP(personalized propagation of neural predictions). Surprisingly, some recent experiments by simply adapting our initialization to SOTA normalization methods illustrate performance benefit (eg. with PairNorm we 16 layer and 32 layer GCN, we observed around ~12.5% and ~33.9% improvement fro Cora), which is highly encouraging.
> >
> > **[Cons5] Additional justification for construction for equation 14?** We obtained Eq. 12 & 13 as the solution to the objective Eq. 11. We have deferred the detailed derivation into Appendix A(supplementary). The key step is to divide Eq. 11 into a sum of block-wise differences. Eq. 12 & 13 can be simply achieved by sampling each weight independently from a distribution with variance $Eq. 12 / C’$. We consider uniform distribution since it has bounded values. Since uniform distribution $U[-a, a]$ has variance $a^2/3$, we can derive the lower/upper limit of the uniform distribution in Eq. 14.
> >
> > We hope our responses have clarified many of your concerns, and please do not hesitate to let us know what else we could do in order to convince you for a rating upgrade.
> >
> > [2] Qi, Haozhi, et al. "Deep isometric learning for visual recognition." International Conference on Machine Learning. PMLR, 2020.
> >
> > [3] Lee, N., Ajanthan, T., Gould, S., & Torr, P.H. (2020). A Signal Propagation Perspective for Pruning Neural Networks at Initialization. ICLR 2020

---

> ### Author Response · Authors · 2022-08-05
> **Author Response for Reviewer ePpt**
>
> Dear Reviewer ePpt,
>
> We thank you for your time to review our work and your constructive comments to improve it, and we really hope to have a further discussion with you to see if our response solves your concerns. We have replied to the important points raised by you such as the cost of DE calculation, application of our methods for MLPs, the relationship between DE and gradient flow, etc in our rebuttal response. Since the author-reviewer discussion period has started for a few days, we will appreciate if you could check our response to your review comments soon. This way, if you have further questions and comments, we can still reply before the author-reviewer discussion period ends.
>
> If our response resolves your concerns, we kindly ask you to consider raising the rating of our work. We again thank you for your time and efforts.
>
> Best, Authors of Paper7865

---

> ### Author Response · Authors · 2022-08-08
> **Author Response for Reviewer ePpt**
>
> Dear Reviewer ePpt,
>
> Since the author-reviewer discussion period is about to get over in the next ~24 hours, we are very eagerly waiting for your response to our rebuttal comments. We sincerely hope that you have read our rebuttal comments, and if you have any further doubts, please let us know as soon as possible so that we can respond to them within the time limits. If our response resolves your concerns, we kindly ask you to consider raising the rating of our work.
>
> Thanks! Authors

---

> ### Comment · Reviewer_ePpt · 2022-08-09
> **Thank you for the reply - scoring unaffected**
>
> I thank the authors for their reply.  Some of my concerns have been addressed. However, I still concern the the computational burden introduced by the update of Dirichlet energy. I'll maintain my score.

---

> > ### Author Response · Authors · 2022-08-09
> > **Author Response for Reviewer ePpt**
> >
> > Dear Reviewer ePpt,
> >
> > We thank you for your response and your time to read our rebuttal.
> >
> > We are glad to know that our rebuttal addressed some of your concerns. As we have explained that considering the real-world graphs are highly sparse, our DE estimation method during the first few epochs of training scales only linearly with the number of nodes in the graph, which is similar to training a GCN layer. To be more precise, practically on our RTX A6000 GPUs, training vanilla 32-layers GCNs with our techniques incorporated (DE and new init) only takes ~2.10%, ~1.97%, ~4.33%, and ~6.04% more time than the plain vanilla GCNs for Cora, Citeseer, Pubmed, and OGBN-ArXiv datasets (Note that these numbers are estimated only for the training duration and we have excluded the model/dataloader creation and loading time).
> >
> > We hope it helps clarify your confusion and concern on the computational burden introduced by the update of Dirichlet energy. If you could kindly check our response and please consider raising the score, that will be deeply appreciated.
> >
> > Thanks,
> > Authors

---

### Official Review · Reviewer_eHHR · 2022-07-13

**Rating:** 3
**Confidence:** 4
**Soundness:** 2 fair
**Presentation:** 2 fair
**Contribution:** 2 fair

**Summary:**

The authors study the gradient flow of deep GCNs and derive a topology-aware isometric initialization scheme for vanilla-GCNs. Then, they propose to use Dirichlet Energy for dynamic rewiring of vanilla-GCNs with skip-connections to increase the expressive power of GCN.


**Questions:**

1).  Line 76, “we leverage Dirichlet Energy (which measures the expressiveness of feature embedding)…” I cannot see why Dirichlet Energy measures the expressiveness of feature embedding?

2). In equation (4), why is the gradient flow defined as the mean of the p-norm of the gradients for all the L layers.

3). What is the standard deviation of the results in table 2.

4). What is the “Dirichlet Energy guided rewiring” looks like? Write it in math.

Writing:

1). The position of Figure 1 need to be adjusted.

2). Figure 3 is messy, I suggest only keeping one set of legend and moving it outside the plot.

3). To simplify the computation, in equation (6), you can write $y_i = WX(A+I)e_i$, where $e_i$ is the column one-hot vector with 1 in I-th element and 0 for others.


**Limitations:**

The authors adequately addressed the limitations and potential negative societal impact of their work

**Strengths And Weaknesses:**

## Strengths:
1. The writing is easy to follow.

2. The experiments are comprehensive.


## Weaknesses:

1. The method and results have some novelty but not significant.

2. Some plots are messy and some arguments are not well support by enough evidence.

3. Some Missing literature on the gradient flow and trainability of GNNs. [1,2]

[1] Luan S, Zhao M, Chang X W, et al. Training matters: Unlocking potentials of deeper graph convolutional neural networks[J]. arXiv preprint arXiv:2008.08838, 2020.

[2] Cong W, Ramezani M, Mahdavi M. On provable benefits of depth in training graph convolutional networks[J]. Advances in Neural Information Processing Systems, 2021, 34: 9936-9949.

---

> ### Author Response · Authors · 2022-08-01
> **Author Response for Reviewer eHHR**
>
> We would like to thank you for your time in reviewing our work and your appreciation for the writing and comprehensiveness of our experiments. We would like to address your concerns one by one below:
>
> **[Cons1] Method and results have some novelty but not significant?** We would like to highlight that our work extends the intuition of "over-smoothening” and "information bottlenecks” which hurt the trainability of deep GCNs and presents a comprehensive study of the training hurdle from the signal propagation perspective and show how improving gradient flow can help in a significant performance gain. All other reviewers (K9SU, ePpt, and 7SM5) have acknowledged our motivation experiments as very insightful and performance gain quite significant. More specifically,
>
> (a) Firstly, our novel isometric initialization technique for GCNs helps in mitigating gradient flow issues and provides significant performance benefits with an increase in depth. Note that, unlike many fancy modern SOTA methods which increase the architectural complexity of GCNs and suffer from memory issues with large graphs, the benefits provided by our method don't come with any architectural overhead and it encourages the GNN community to reconsider initialization schemes of GNNs.
>
> (b) Secondly, our DE-guided rewiring of GCNs is an interesting and insightful idea (acknowledged by K9SU - “impressed by dynamic skips”), which introduces skip connections only when a layer requires it. For example, static skips introduced in [jumping, residual, etc…], by default incorporate skips for training shallow GCNs with 2-3 layers (which do not require it), and tend to hurt the final performance. For example, in table 3, it can be observed that training a 2-layered GCN with initial, jumping, or residual skips suffers -2.1%, -0.12%, -6.37% performance drop compared to a plain vanilla GCN (81.10%) on Cora. Our method only introduces skips when it is required by tracking the DE of the layer, and thereby overcomes the issue of adding skips in shallow GCNs, and for layers that receive healthy gradient flow. Our principled way of introducing skip connection can easily beat the performance benefits of many SOTA skipping schemes which come up with additional computational overheads, as well as many normalization, and random dropping methods.
>
> **[Cons2] Plots are messy and some arguments are not well support by enough evidence?** Thanks for your suggestions on Figure 1 and Figure 3 and we have updated them in the revised version. Regarding evidence support, we would try our best to address your concerns if you can explicitly point out the arguments which require additional evidence. Without that, it is ambiguous and impossible for us to locate those arguments.
>
> **[Cons3] Missing Literature?** We would like to thank you for pointing out some missing relevant literature and we have incorporated their discussion in our revised draft in the introduction section. Moreover, comparing the best numbers reported by [1] on Cora test set after incorporating their techniques combination for a 10-layer GCN is  73.98 $\pm$ 2.68%(Table 1) which is significantly lower than our method (82.93 $\pm$ 0.21%) with the similar settings.
>
> **[Cons4] Why Dirichlet Energy measures the expressiveness if the feature embedding?** We would like to highlight that Dirichlet energy usage to measure the expressiveness of feature embedding has been established by Cai et. al 2020 (ICML 2020)[2]. It provides detailed insights that Dirichlet energy decreases exponentially with respect to the number of layers, and the feature embeddings start losing discriminative/expressive power. We encourage you to look at the paper for more details.

---

> > ### Author Response · Authors · 2022-08-01
> > **Author Response for Reviewer eHHR**
> >
> > **[Cons4] Why gradient flow defined as the mean of p-norm of the gradients?** We would like to highlight that traditional measure to estimate gradient flow in DNNs has been established as p-norm of the gradients (Terressa et. al.2021, Chen et al., 2018; Pascanu et al., 2013; Evci et al., 2020), and our estimation of average gradient flow across different layers is conformed to previous works. Moreover, all reviewers have accepted its validity and reviewer K9SU and ePpt explicitly acknowledge our gradient flow measurement and observations are useful, interesting, and insightful.
> >
> > **[Cons5] Mathematical representation of Dirichlet Energy Guided rewiring?** Thanks for this suggestion. Mathematically, our dynamic rewiring can be written as $X^{l} =\mathbb{1}{[E(X^{(l_t)}) < pE(X^{(l_0)})]} \alpha X^{(1)} + \tilde{X}^{l}$
> > where $\mathbb{1}[\cdot]$ is an indicator function, $t$ denotes the training epoch, $p$ denotes energy drop threshold, $\alpha$ denotes skip information ratio, and $\tilde{X}^{l} = W_{l-1}X^{(l-1)}(A+I)$. We have added this formula in our revision.
> >
> > [1] Luan S, Zhao M, Chang X W, et al. Training matters: Unlocking potentials of deeper graph convolutional neural networks[J].
> >
> > [2] Cai, Chen, and Yusu Wang. "A note on over-smoothing for graph neural networks." arXiv preprint arXiv:2006.13318 (2020).
> >
> > We hope our responses have clarified many of your concerns, and please do not hesitate to let us know what else we could do in order to convince you for a rating upgrade.

---

> > > ### Comment · Reviewer_eHHR · 2022-08-09
> > > **Response to Authors**
> > >
> > > Thanks for your response and some of my questions are addressed, but I still have some concerns left:
> > >
> > > 1. Comparison with the results in [1]: You should compare with the results in table 2 of [1], not table 1. I don't see significant improvement.
> > >
> > > 2. Dirichlet Energy and the expressiveness: From my point of view, using Dirichlet Energy to measure the expressiveness of the feature embedding is only valid for homophilic graphs, where the smoothness defined by graph Laplacian is valid. For heterophilic graph, this smoothness is invalid. Thus, this measurement is not generally effective. (By the way, [2] is not a published ICML 2020 paper, please check)
> > >
> > > 3.  About the mean of p-norm of the gradients: It's better to explain it directly and I didn't find the papers (Terressa et. al.2021, Chen et al., 2018; Pascanu et al., 2013; Evci et al., 2020) in your reference.
> > >
> > > Suggestion: Other reviewers acknowledged your work does not mean I need to acknowledge as well. I suggest answering my question directly and clearly instead of refering to others. This is not a good strategy for your rebuttal.
> > >
> > > Based on the above arguments, I still have concerns on the effectiveness of the weight initialization method and Dirichlet Energy Guided rewiring method. Thus, I'll keep my rating.

---

> ### Author Response · Authors · 2022-08-05
> **Author Response for Reviewer eHHR**
>
> Dear Reviewer eHHR,
>
> We thank you for your time to review our work and your constructive comments to improve it, and we really hope to have a further discussion with you to see if our response solves your concerns. We have replied to the important points raised by you such as how DE measures expressiveness, novelty concerns, and missing mathematical equations for the DE-guided rewiring, etc in our rebuttal response. Since the author-reviewer discussion period has started for a few days, we will appreciate if you could check our response to your review comments soon. This way, if you have further questions and comments, we can still reply before the author-reviewer discussion period ends.
>
> If our response resolves your concerns, we kindly ask you to consider raising the rating of our work. We again thank you for your time and efforts.
>
> Best,
> Authors of Paper7865

---

> ### Author Response · Authors · 2022-08-08
> **Author Response for Reviewer eHHR**
>
> Dear Reviewer eHHR,
>
> Since the author-reviewer discussion period is about to get over in the next ~24 hours, we are very eagerly waiting for your response to our rebuttal comments. We sincerely hope that you have read our rebuttal comments, and if you have any further doubts, please let us know as soon as possible so that we can respond to them within the time limits. If our response resolves your concerns, we kindly ask you to consider raising the rating of our work.
>
> Thanks!
> Authors

---

### Meta-Review · Area_Chair_R8MR · 2022-08-26

**Recommendation:** Accept
**Confidence:** Less certain

**Metareview:**

The paper proposes to promote the training of deep GCNs from the gradient flow perspective by introducing topology-aware isometric initialization and Dirichlet energy guided rewiring. The intuition of looking into healthy gradient flow is innovative and the observations are insightful. The experiments generally perform good, which demonstrate the effectiveness of proposed method. The reviewers have raised several concerns about the technical and experiments details, such as the use of Dirichlet energy and the discussion of trainability issue. The authors provide a nice rebuttal, and the discussions should be included in the revision to better present the research work.

**Award:**

No

---

### Decision · Program_Chairs · 2022-09-14

Accept